# Session RPE Breakpoints Corresponding to Intensity Thresholds in Elite Open Water Swimmers

**DOI:** 10.3390/jfmk5010021

**Published:** 2020-03-17

**Authors:** Cristian Ieno, Roberto Baldassarre, Claudio Quagliarotti, Marco Bonifazi, Maria Francesca Piacentini

**Affiliations:** 1University of Rome Foro Italico, 00135 Rome, Italy; c.ieno@studenti.uniroma4.it (C.I.); roberto.baldassare@me.com (R.B.); c.quagliarotti@studenti.uniroma4.it (C.Q.); 2Italian Swimming Federation 2, 00135 Rome, Italy; marco.bonifazi@unisi.it; 3Department of Medical, Surgical and Neuro Sciences, University of Siena, 53100 Siena, Italy

**Keywords:** endurance, polarized training, TL, training zones, rate of perceived exertion, lactate thresholds detection, elite athletes

## Abstract

This study aims to assess the correspondence between session rating of perceived exertion (sRPE) breakpoints with both the first lactate threshold (LT1) and the second lactate threshold (LT2) in elite open water swimmers (OWS). Six elite OWS of the National Olympic Team specialized in distances between 5 and 25 km participated to the study. OWS performed a set of 6 times 500 m incremental swimming step test during which blood lactate concentration (BLC), split time (ST), stroke frequency (SF), and rating of perceived exertion (RPE) were collected. To assess the corresponding breakpoints, we considered LT1 as the highest workload not associated with rise in BLC and LT2 as the increase of 2mM above LT1. According to the LT1 and LT2, the identified zones were: Z1 ≤3, Z2 between 4 and 6, Z3 ≥ 7. In conclusion, the intensity zones determined for OWS resulted different from what previously reported for other endurance disciplines.

## 1. Introduction

Training load (TL) monitoring helps coaches modulate load in order to guide periodization, guarantee maximal recovery, and avoid negative effects such as non-functional overreaching [1,2]. Specifically, elite athletes incorporate periods of functional overreaching in a periodized training plan, and progression towards non-functional overreaching or overtraining is undesirable [3]. Therefore, it is imperative to be able to monitor correctly individual response to training and for this reason, training monitoring has been one of the most studied topics in the past years [3,4,5]. Both objective and subjective methods have been studied and compared [3], however, subjective methods such as online training diaries [1], athlete well-being [3], or perceived fatigue are generally the most utilized. The use of session rating of perceived exertion (sRPE) as a method to quantify internal TL has been validated for different sports [6,7,8,9] including swimming [10,11]. Despite the large amount of research performed on training of elite endurance athletes of different disciplines, there is a lack of studies specifically reporting data regarding training of elite endurance open water swimmers (OWS), considering the diversity of the discipline and the high technical requirement necessary to succeed [12]. Nevertheless, OWS have different physiological characteristics compared to pool swimmers, dictated by the different energy demands of their competitions [13]. Indeed OWS have a head to head competition (compared to lane based pool swimmers) and compete over long distances (5, 10, and 25 km) often in cold water; this requires OWS to have a great aerobic capacity and a different percentage of fat mass compared to pool swimmers [12,14].

Recently, Seiler and Kjerland [15], in the attempt of quantifying training intensity distribution in cross country skiers, compared three different independent monitoring systems, heart rate (HR), blood lactate concentration (BLC), and sRPE. Specifically, the modified 10-point scale developed by Foster and colleagues [2,6,16] was divided in three Zones (Z1, Z2, and Z3) according to the first (VT1) and the second ventilatory threshold (VT2) allocating in Z1 values reported for each session (sRPE) below or equal to four (below VT1), in Z2 values of five and six (between VT1 and VT2), and in Z3 values equal to or above seven (and above VT2) [15]. sRPE would seem the easiest and most reliable [11] method to monitoring TL in swimmers, as reported by Baldassarre and colleagues [17], because of the objective difficulty in using technological support in aquatic environments. However, Wallace et al. [10] quantified TL distribution in swimming with sRPE, by dividing the rating of perceived exertion (RPE) scale (CR10) [18] into three zones: “easy” as RPE 1 and 2, moderate (RPE 3-5) and hard (RPE> 5) without reporting how they allocated these specific breakpoints. Considering that in swimming the energy cost increases exponentially with increases in speed [19,20] we hypothesized that the breakpoints for “terrestrial” [15] endurance sports would not be applicable to endurance swimming [21]. Given the objective difficulties in estimating ventilatory thresholds and that blood lactate concentration has been shown to have a high correlation with RPE [11], the aim of the present study was to evaluate the correspondence between sRPE breakpoints with the first (LT1) and the second lactate threshold (LT2) in a group of elite OWS.

## 2. Materials and Methods 

### 2.1. Subjects

Six elite OWS (four females and two males; 26 ± 2 yrs, 176.5 ± 9.95 cm, 66.25 ± 11.62 Kg) all part of the National Olympic Team, participated in the study. Athletes were specialized in distances between 5 and 25 Km and were World or European champions and Olympic medalists. All swimmers were informed about the procedures of the study and written informed consent was obtained from the subjects participating in the study. The Institutional review Board of the University of Rome “Foro Italico” approved the study (CAR 02/2019) and the data were collected in collaboration with the national swimming federation. Data collection was performed between October 2018 and December 2019 on four different days for a total of 13 sessions. Every athlete participated at all experimental sessions.

### 2.2. Experimental Procedure: Incremental Exercise Test

Immediately after 20-min of a standardized warm-up at self-selected swimming speed (~1.30 m·s^−1^), swimmers performed a set of 6 times 500 m swimming incremental step test that is the typical test performed by these athletes to monitor their performance progress throughout the year. The test was performed in a 50 m long course swimming pool. The ramp intensity was selected by the coach according to the fitness level of each swimmer (progressive increase based on their personal best over the distance) [22]. Participants were instructed to increment the speed in each step in order to perform the last 500m near maximal effort [22]. The speed increments are shown in Table 1. At the end of each 500m, the athletes were stopped only for the time needed for lactate measurement. Specialized and authorized medical personnel collected the blood samples. BLC was analyzed with the Lactate Pro System 2 (Kyoto, Japan). During the sampling at the end of every step athletes were asked to rate their perceived exertion through the modified 10-point scale by Foster [2,6,16].

During the test, step times were measured through a stopwatch (FINIS 3X-300M, FINIS, Inc., Livermore, CA, USA) and stroke frequency (SF) was measured on the third lap by timing three complete stroke cycles [23]. All swimmers were fully familiarized with the RPE scale and all the instrumentation utilized during the test.

### 2.3. Data Analysis

The RPE, lactate, and speed values collected during the step tests were interpolated through a third-degree polynomial in order to determine the estimated lactate and speed values for every RPE scale point (Table 2 and Table 3). Regarding the speed maintained during each step, averages were calculated separately for women and men (Table 3). 

LTs were calculated as described by Pallarés [24], where the highest workload not associated with a rise in BLC above baseline was considered LT1 and a rise of 2mM above LT1 was considered as LT2. The LTs were used to delineate the intensity zones (three zone model) (Table 2). 

Descriptive statistics are presented as mean ± standard deviation (SD).

## 3. Results

### Intensity Zones

The mean speed, RPE, BLC, and SF for each step are shown in Table 1. Table 2 shows the calculated breakpoints on the RPE scale, delimiting the three intensity zones, useful for monitoring training load by sRPE. 

Swimming speed based on the step test corresponding to each value of the RPE scale is reported in Table 3.

## 4. Discussion

This study aims to verify that the correspondence between the breakpoints of the sRPE scale developed by Foster [2,6,16] is useful for determining the three training intensity zones, with the LTs in elite OWS in order to correctly monitor and quantify TL distribution. 

The main finding of our study was that the breakpoints are differently allocated than what previously reported for other endurance disciplines and Z2 includes more values than previously reported [15]. 

Organization of the training continuum in intensity zones is common within the coaching community, with the zones often defined by physiological parameters (HR or BLC) [15], and also if these intensity boundaries are not clearly defined. Nevertheless, recent work [25,26] clearly showed that anchoring the three intensity zones to first and second VT clearly define time to fatigue and stress load of each of the three zones defining Z1 as a session performed below VT1, Z2 as a session performed between VT1 and VT2 and Z3 a session performed at or above VT2. 

Seiler and Kjerland [15] compared three different independent monitoring systems, HR, BLC, and sRPE with the “session goal approach” therefore allocating the whole session to one specific zone. In particular, HR and sRPE were distributed in the three intensity zones, outlined by the VTs. Similarly, a whole session was allocated in Z1 for values below 2 mM of BLC, in Z2 for values between 2 and 4 mM, and in Zone for values below 4 mM [15]. The authors found a good agreement between all three methods, specifying that the sRPE breakpoints corresponding to ventilatory thresholds are a good and practical method to quantify training intensity distribution on three intensity zones. Although physiological parameters of OWS are comparable to those of athletes competing in different endurance sports [17], LT2 occurs at lower blood lactate concentrations.

Compared to what previously reported [10,15], our Z1 excludes “4” that is positioned in Z2. Although we measured lactate and not ventilatory parameters because of the aquatic environment, Lucia et al. [25] found a good correspondence between the two that can be used interchangeably.

We hypothesized that Z2 would be larger to what previously reported, because of the energy cost of water locomotion. In fact, the capability to transform the mechanical power produced by the muscles, known as propelling efficiency, into “useful” power to move in the water is one of the most important limiting factors in swimming performance.

In fact, as described by Di Prampero [19] the speed depends on the energy cost: Vmax = FVO2max/C(1)
where C is the energy cost to cover one-unit distance at a given specific mode of locomotion speed, which in the case of swimming (CS) depends of drag, or hydrodynamic resistance (Wd), propelling efficiency (ηp), and overall efficiency (ηo): CS = Wd/(ηp × ηo)(2)

Considering that Wd increase exponentially with increasing speed, a minimum increase of the latter will determine a substantial increase in energy cost (and fatigue). 

Performance improvements in swimming are more likely accounted for by improved technical ability [20,27]. Therefore, both the specific requirements of OWS (competing at or near lactate threshold [14]) and the physical properties of the water (effect of drag) explain the higher percentage of training in Z2 compared to other endurance events as shown by Baldassarre et al. [17]. The extreme event in aquatic environments that are performed need a different training intensity distribution compared to land endurance athletes. Training at threshold or in Z3 may compromise the efficiency of the technique and stroke mechanism [20]. In fact, a high volume at low intensity training is normally utilized by coaches to increase aquatic skills of their athletes as the efficiency of locomotion is the parameter that needs to be maximized [12,20]. 

To sustain high swimming speed for many hours, OWS should be able to sustain a high percentage of V̇O2max (80–90%) for many hours [12]. The mean speed of the best OWS during the 10 km race for women and men have been reported to be, respectively, 1.39 m·s^−1^ and 1.44 m·s^−1^ [13] that both correspond at values below LT1 (Table 3). However, the best male and female OWS normally adopt a negative pacing [13,28], increasing significantly the speed in the last few hundred meters of the race. In fact, in the first lap of the race (2.5 km) the male medalists are allocated around the 40th–57th place and then progress towards the 8th place in the last lap while the female medalists are allocated around the 19th–30th place in the first lap and thereafter progress towards the first positions in the last lap [28]. Being sheltered in the group allows swimmers to draft, which determines the more conservative approach of the fastest swimmers during the first parts of the race. Swimming behind another swimmer at a distance of 0 to 50 cm reduces by 11%–38% the metabolic response of the draftee [29]. In fact, in the last part of the race, athletes significantly increase their speed (1.56 m·s^−1^ for males and 1.46 m·s^−1^ for females) corresponding to speeds above LT2 for males while the women’s speed over the last split is around the first lactate threshold. Specifically, males reach speeds corresponding to or above the LT2 because of the higher performance density of their races (i.e., the difference between the first and the 10th male during the Olympic Games in Rio was only 0.07%) [12]. This is in fact confirmed by the fact that the best OWS are also the fastest in shorter pool events compared to lower level OWS, confirming that a high anaerobic speed reserve is necessary for these athletes [13].

Considering that monitoring training by sRPE is a validated method that allows coaches to understand if the response of the athlete corresponds to what was prescribed [10,30], even in expert swimmers [31], the establishment of the correct breakpoints on the sRPE scale corresponding to correct physiological breakpoints is a fundamental step to allocate the training session in an accurate way. In addition, monitoring internal TL with maximum accuracy is essential in avoiding and preventing non-functional overreaching or overtraining. 

## 5. Study Limitation

The major limitation in the present study is small sample size. Therefore, these breakpoints are specific for the elite athletes tested and could be applicable to athletes of the same performance level, however more research is necessary. Moreover, a difference between males and females could not be established.

## 6. Conclusions

sRPE breakpoints for OWS were different compared to sports that adopt a different mode of locomotion, probably due to the different energy cost of locomotion in the water. When prescribing an aerobic type of workout it is essential to take into consideration the work time performed in each zone [32]. It is already well known that in swimming, perception of effort is influenced not only by intensity but also by volume and repetition distance [33]. OWS coaches often prescribe many km around LT1 that they would consider in Z1. However, as training volume increases, sRPE can easily drift to four that for this population of athletes, is already a workout in Zone 2. The major risk is therefore an unwanted accumulation of training in Zone 2 instead of Zone 1. These findings offer to coaches, physical trainers, and specialists an easy to use tool to monitor the training session of elite endurance open water swimmers. Moreover, also during the race where the use of devices such as a heart rate monitors is not permitted, the athlete’s feedback becomes essential.

## Figures and Tables

**Table 1 jfmk-05-00021-t001:** Mean speed, rating of perceived exertion (RPE), blood lactate concentration (BLC), and stroke frequency (SF) for each step of the incremental tests.

6 × 500 Step Test				
Step	Speed (m·s^−1^)	RPE	BLC (mM)	SF (Cycles·min^−1^)
**I**	1.33 ± 0.04	0.6 ± 0.3	1.4 ± 0.4	32 ± 3
**II**	1.40 ± 0.03	1.8 ± 0.4	1.2 ± 0.3	34 ± 3
**III**	1.44 ± 0.03	3.0 ± 0.7	1.3 ± 0.3	37 ± 3
**IV**	1.47 ± 0.04	4.3 ± 1.1	1.8 ± 0.9	39 ± 3
**V**	1.50 ± 0.04	6.0 ± 1.3	3.0 ± 1.5	41 ± 4
**VI**	1.54 ± 0.05	8.9 ± 1.4	6.1 ± 1.8	44 ± 5

**Table 2 jfmk-05-00021-t002:** Interpolated breakpoints on the session rating of perceived exertion (sRPE) scale in open water swimmers (OWS) corresponding to first rise/+2 mM of BLC.

	0—Rest		
	1—Very Easy	1.3	
	2—Easy	1.2	**Z1**
LT1	3—Moderate	1.3	
	4—Somewhat Hard	1.7	
	5—Hard	2.2	**Z2**
LT2	6	3.0	
	7—Very Hard	4.0	
	8—Very, Very Hard	5.2	**Z3**
	9—Nearly Maximal	6.8	
	10—Maximal Effort	8.6	

**Table 3 jfmk-05-00021-t003:** Interpolated mean speed for women and men in relationship with breakpoints.

sRPE	Speed (m·s^−1^)	
	Women	Men
0—Rest		
1—Very Easy	1.37 ± 0.03	1.39 ± 0.01
2—Easy	1.41 ± 0.04	1.44 ± 0.00
3—Moderate	1.44 ± 0.04	1.47 ± 0.01
4—Somewhat Hard	1.46 ± 0.03	1.50 ± 0.03
5—Hard	1.47 ± 0.03	1.52 ± 0.05
6	1.48 ± 0.02	1.54 ± 0.06
7—Very Hard	1.49 ± 0.03	1.55 ± 0.06
8—Very, Very Hard	1.51 ± 0.04	1.57 ± 0.03
9—Nearly Maximal	1.54 ± 0.07	1.59 ± 0.05
10—Maximal Effort	1.57 ± 0.14	1.61 ± 0.15

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
