# Peer review of "Session RPE Breakpoints Corresponding to Intensity Thresholds in Elite Open Water Swimmers"

_jfmk, 2020, doi:10.3390/jfmk5010021_

Round 1

Reviewer 1 Report

General comments:

This study examines the “Session RPE breakpoints Corresponding to Intensity 2 Thresholds in Elite Open Water Swimmers”. The subject is very interesting. In general, the study is simple, and very interesting. This being said, the paper need improvement to be published in a scientific journal. Among others, the language use e.g., using velocity and speed interchangeably very often, using sRPE and RPE. The method section is not very well written and need further details regarding testing time points etc… well done and thought study. To improve the presentation of the study here are some specific comments that would improve the paper.

Specific comment:

Introduction

Page

Line

Comment

1

30

Provide reference after the first sentence and before the DOT.

Materials and Methods

2

65

does the ethical approval have a number? please provide if so. if not, please explain in your response.

2

70

Provide a full description of the warm up used.

2

70

Change velocity to speed (this apply to the entire manuscript.

2

70

6x500 m, could the authors provide more details? was it 6 set x 500 m? was there any recovery period between sets? or was it continues 3000 m? did the athletes stopped for measures?

2

73

More explanation is needed regarding how the researchers controlled the increase in swimming speed during the test.

2

76-77

stopwatch (FINIS 3X-300M, 76 Finisswimm): provide, city, country.

what is the reliability and validity of the FINIS system? as reported by the manufacture if the authors did not assess it prior to investigation?

2

77

BLC (Lactate Pro System 2):

provide city and country.

how was the lactate measured? what was the procedure?

2

77

How was the RPE collected? provide a detailed procedure.

2

82

The text should be consequential. use the same wording throughout the text is much better presentation. (Srpe / rpe?)

Results:

2

93

Authors should use on wording, either speed or velocity. I would suggest speed is much better word in this context. this should be corrected through out the whole text.

3

Table 1

Table 1, use the correct measuring si unite for lactate.

4

FIGURE 1

how many time point measurements did the authors measure? was it 6 times or 10 times? further explanation could be provided in the method section?

Discussion:

4

116

(sRPE) there is a lack in the methods where the authors do not provide explanation of how the RPE was collected. furthermore, did the authors collect the sRPE or RPE? as comparing the results with the discussion indicate a conflict.

4

138-144

suggest moving this part to introduction. avoid repeated text.

In general, the discussion should be rewritten where the main focus would be addressing the results of the current study. Therefore, while there are many of the information are useful, many is irrelevant. Its important to avoid repeated text from introduction. Maintain those information that address your results and remove all other moments that are not related. In this section that authors are required to discuss their results and not provide a new introduction.

6

196

Provide a heading prior to conclusion with the limitations of the study.

Reviewer 2 Report

The authors made an important and excellent job when analyzing the potential of RPE to discriminate intensity training zones in open water swimmers (OWS).

Although the small sample size, one of the main strong aspects of the manuscript is the high competitive level of the sample.

The introduction and discussion sections are very well structured and clear point the perspective of the authors. The emerging potential of the use of the RPE to the control training in OWS, highlighting the needs of rigorous criteria to define the cutoff points in RPE in each training zone.

Also, the authors claim to the ecological effect of swimming when compared to other types of terrestrial sports related to the exponential increase in the energetic cost due to the drag, which in turn strongly affects the RPE.

However, In my opinion, I felt the lack of a more clear comparison between the cutoff RPE values determined through the FBLC and that propose by Pallares for LT1 and LT2.

My suggestion is to conduct for each incremental test, and from the two strategies adopted, a pairwise comparison for the cutoff values according to each training zones.

Minor suggestions and english revision (ex. change  "The aim of this study" by “This study aims to…”. Rewrite  “élite”  as "elite" without the accent.

In the methods section, could the authors be more precise in relation to the increment of the intensity in the incremental protocol and add the rest interval between repetitions. 

When the authors state that the study was approved by the local Ethics commission, please be precise (ex. University of Rome?)

Line 78 authors report that RPE was collected at baseline. Could you please add the reason? 

Round 2

Reviewer 1 Report

Dear Authors,

Thank you for your revised version. however, the only concern that I have is the information provided at the method section. the method section should be build so other reseachers can test it (replicability).

Immediately after 20-min of a standardized warm-up at self-selected speed (~1,30 m·s¯¹). what was the warm up? was it swimming? jogging? detailed warm up should be provided.

why was the stroke frequency (SF) calculated? I did not notice any results regarding stroke frequency (SF) at the result section?

how was the stroke frequency (SF) manually calculated?

study limitations: I dont belive that with 6 participants the authors can generalize the results to even "this level of athletes". therefore, it should be clearly stated that the results of this study is valid for those who participated in this study, and further research is needed to be able to generalize to other athletes.
